# Sustainable innovation in the context of organizational cultural diversity: The role of cultural intelligence and knowledge sharing

**Jinlong Li** \*, **Na Wu, Shengxu Xiong**

School of Business Administration, Zhongnan University of Economics and Law, Wuhan, Hubei, China

\* starlijinlong@163.com

## Abstract

With the in-depth development of globalization, individuals are increasingly embedded in a culturally diverse environment. Effective communication and management ability (Cultural Intelligence) of employees in this type of diverse and heterogeneous environment impacts behavior and performance, affecting the sustainable innovation ability of organizations. Researchers have not yet fully assessed the impact of individuals' cross-cultural management ability on sustainable innovation. Using Cultural Intelligence Theory and Trait Activation Theory, this paper discusses the influence of individual cultural intelligence on sustainable innovation behavior. The results showed that employees' cultural intelligence positively affected their sustainable innovation behavior. Employee knowledge sharing plays an mediating role between intelligence and behavior. Differences in organizational culture have a negative moderating effect on the impact of employees' cultural intelligence on knowledge sharing and sustainable innovation behaviors. The research results provide theoretical guidance for managing organizational cultural diversity and advancing cultural intelligence and sustainable innovation behaviors among employees.

## Introduction

Aggravated by the global spread of COVID-19, cross-cultural conflicts such as trade barriers, ultra-nationalism, and regional discrimination have increasingly intensified worldwide, highlighting the challenges faced in a multicultural organizational environment [1, 2]. At the same time, the global flow of human resources and the influx of a large number of "new generation employees" have led to employee differentiation, resulting the "personalized" development of corporate culture [3]. The "personalization" of multicultural organization environments and corporate culture requires employees to engage with groups form different cultural backgrounds, and cross-cultural communication has become the norm in employee activities. Cultural differences and cultural barriers have become important factors affecting employee behavior and performance, posing challenges for the innovative management, and sustainable development of enterprises [2, 4]. This context highlights the importance of cross-

**Data Availability Statement:** All relevant data are within the manuscript and its Supporting Information files.

**Funding:** This study was supported by the National Social Science Foundation of China to XS (Project

No. 15BGL036) and the Innovation Program of Zhongnan University of Economics and Law to LJ (Project No. 201911049).

**Competing interests:** The authors have declared that no competing interests exist.

cultural communication and the ability to manage employees, making it worthy of in-depth study in academia and management practice.

Individual sustainable innovation behavior has had an increasingly important influence on the sustainable development of enterprises. The innovation behavior of individuals in an organization forms a micro foundation of enterprise innovation and sustainable development [5]. Studies have focused on three main factors affecting individual innovation [6]. This first focuses on the individual characteristics of employees in the organization, such as personality, psychology, self-efficacy, knowledge, and ability [7–9]. The second focuses on job characteristics, including task complexity, work challenge, job autonomy, feedback, and rewards [10, 11]. The third focuses on organizational factors, including organizational leadership, organizational atmosphere, organizational system, human resource management, and organizational culture [5, 6]. Individual knowledge and ability are key factors affecting innovation behavior, however, individual sustainable innovation is affected by organizational environment factors, such as organizational culture, diversity of organizational culture, and the atmosphere created by organizational cultural differences [6].

Organizational culture is considered a driving force for innovation, providing enterprises with lasting competitive advantages and serves as a key factor for sustainable development [12, 13]. Many scholars have studied the kinds of organizational culture that promote innovation and the kinds of culture that lead to an enterprise's sustainable development (such as learning culture, open culture, green culture, etc.) [13–15]. These studies have not focused on the impact of cultural diversity management within organizations and the impact of organizational cultural differences on innovation and sustainable development. Rather, research on organizational cultural diversity has focused on the impact of cultural diversity and organizational cultural differences on economic effects and sustainable development at the organizational level [16, 17]. At the individual level, researchers have focused on the impact of cultural diversity characteristics on employees' work performance and adaptability [18, 19]. However, the impact of individual employee cultural diversity management ability on sustainable innovation has not received corresponding attention. This is particularly true in organizational environments, where the cultural differences are increasingly visible due to the diversity of organizational culture. As such, the factors influencing employees' sustainable innovation behavior deserve more attention.

An employee's ability to manage cultural diversity is a part of individual ability that deserves special attention. To explore the reality that some people communicate and perform better than others in the context of cultural diversity, Earley and Ang (2003) proposed the concept of "Cultural Intelligence" (CQ), defining it as an individual's ability to play a role and effectively manage in the context of cross-cultural environment or cultural diversity [20]. Scholarly studies have shown that individuals with higher cultural intelligence have higher job performance with respect to cross-cultural adaptation, cross-cultural communication, and job adjustment [21, 22]. CQ also impacts an individual psychological well-being, interpersonal trust, and knowledge interaction [23, 24]. Therefore, as a unique index of individual ability and as an indicator of cross-cultural interaction ability, cultural intelligence has received more attention by scholars. However, few studies have examined how cultural intelligence affects employees' sustained innovation behavior, leading to a lack of understanding of this area.

Knowledge sharing is defined here as individual behaviors that help others or cooperate with others to solve problems and conceive new ideas by sharing information and knowledge [25]. Factors influencing knowledge sharing include subjective factors, knowledge characteristics, organizational factors, and situational factors. Knowledge sharing impacts the output, competitiveness, and performance of individuals, teams, and organizations [26–28]. Some studies have demonstrated the impact of knowledge sharing on employees' innovation

behaviors, particularly focusing on its bridging role as an intermediary factor [29, 30]. This study focuses on the issue of knowledge sharing from the perspective of cultural diversity, and examines the influence of individual CQ on knowledge sharing behavior and willingness. It also examines the issue of knowledge sharing among employees when encountering changes in organizational culture differences.

Compared with the existing research, this study provides new insights about the sustainable innovation behavior of employees in the context of organizational cultural diversity. First, previous studies on individual innovation behavior have not considered the impact of employee diversity cultural management ability (CQ). The impact of employee cultural intelligence, as an individual's cross-cultural ability and trait, on an employee's sustainable innovation behavior deserves further study. Second, previous studies on the impact of organizational culture on employees' innovation behavior have considered specific aspects, but have not integrated cross-cultural ability, the organizational culture situation, and knowledge interaction. Third, although the impact of knowledge sharing on employee innovation behavior has been empirically verified, more research is needed to assess whether employees are willing to share knowledge with colleague with different cultural backgrounds in the context of organizational cultural diversity.

Therefore, based on the background of organizational cultural diversity, this study applied an empirical analysis to investigate the internal mechanism and boundary conditions of the impact of employee cultural intelligence on employees' sustainable innovation behavior. It also explored the mediating role of knowledge sharing and the moderating role of the degree of organizational cultural difference. This study integrated individual competence, organizational context, and knowledge interaction factors into an integrated model that considers the sustainable innovation behaviors of employees in the organization. Study conclusions provide a theoretical basis for enterprises to pay attention to employees' cultural intelligence to facilitate employees' continuous innovation and realize the sustainable development of enterprises. The research related to organizational cultural diversity is also expanded at the individual level.

## Theoretical background and hypotheses

### Cultural intelligence and sustainable innovative behavior

**Cultural intelligence.**    Based on the Theory of Multiple Intelligence (Gardner, 1983) and Sternberg and Detterman's (1986) multiple-loci framework of intelligences [31, 32], Earley and Ang (2003) developed the concept of "Cultural Intelligence" (CQ), which assesses an individual's ability to effectively adapt to the new cultural environment [20]. Ang et al. (2007) further improved the definition: CQ reflects an individual's capability to function and manage effectively in a culturally diverse environment. CQ is a specific form of intelligence, focused on the capability to grasp, reason, and behave effectively in situations characterized by cultural diversity [33]. After the concept of CQ was proposed, many scholars studied the connotation, structural dimension, influencing factors, and influencing effects of CQ, forming a systematic Theory of Cultural Intelligence [21, 34, 35]. CQ is widely used in management, psychology, education, and other fields to measure an individual's cross-cultural adaptability, communication level, task performance, and innovation ability [36–39]. With the deepening of global economic integration, cross-border communication has become the norm; employee expatriation and cross-cultural activities have become increasingly frequent; innovation has increasingly become a transnational and trans-regional cooperation model. Given this background, the study of employee cultural intelligence is an important and meaningful subject.

**Sustainable innovation behavior of employees.**    Most scholars have found that the sustainable development of organizations requires innovation [40]. The study of sustainable

innovation is a research problem that has been, and will likely remain, a current issue. Human resource management, technology research and development, environmental protection, and market development are current hot topics in the sustainable innovation of organizations [41, 42]. In the field of organizational management, researchers focus on the impact of enterprise strategy and organizational leadership on sustainable innovation [43, 44], but have not fully explored the sustainable innovation behavior of grassroots employees who are the front-line employees in the workplace. However, the continuous innovation of individuals is the foundation of the sustainable development of the organization. Focusing on the innovation behavior of grassroots employees in the workplace may support development opportunities for the organization; similarly, the sustainable innovation of employees requires organizational support. The sustainable innovation of employees requires the support of the organizational system, a supportive organizational culture, and the creative ability and characteristics of employees themselves [19].

Scott and Bruce (1994) viewed innovation as a multistage process, which is a collection of actions taken by individuals to seek, develop, and apply new ideas and solutions in their current situation [45]. Innovation requires divergent thinking, understanding, and looking at things from different perspectives; and then synthesizing unrelated processes, products, and materials into something new and better [46]. Employees' innovation behaviors are expressed in the form of opinion statements, experiments, and output; the organization converts these into transformative actions to create a competitive advantage [10]. Amabile and Pratt (2016) studied the innovation dynamic model in organizations and noted that individual innovation behavior is affected by many factors, including individual ability and motivation, organizational environment and leadership characteristics, and interactions between individuals and the organizational environment [47].

Some scholars have studied employee creativity, which usually refers to the generation of novel and valuable ideas, and which is theoretically similar to innovation behavior. Pieterse (2010) proposed that individual innovative behavior and creativity are two different concepts [48]; the purpose of creativity is to produce novel and useful ideas, whereas innovation behavior mainly refers to the act of generating new ideas and translating those ideas into practice. Creativity is considered as the first step of innovation and is key to forming sustainable innovation behavior [46]. This study does not specifically distinguish creativity from innovation behavior, and assumes they are similar.

**The influence of cultural intelligence on sustainable innovation behavior.** Tett and Burnett (2003) proposed Trait Activation Theory [49], which refers to the process of an individual expressing his or her characteristics when faced with situational clues related to personality. Trait Activation Theory clarifies how the external situation activates an individual's internal traits and explore the organic connection between internal traits and external situations [50]. CQ focuses on the capability of individuals to effectively manage in the context of cultural diversity [21]; this is similar to Trait Activation Theory, which focuses on the influence of the "person-environment" interaction on individual behavior. Employees' innovation behavior is influenced by personal traits, and is reflected in personal ability, psychological motivation, and behavior performance. CQ is a manifestation of personal abilities and traits. Therefore, in culturally diverse organizations, employees' cross-cultural communication and management ability will impact innovation behaviors. Pandey et al. (2019) studied the impact of CQ on employees' cross-cultural adaptation and work adjustment [51]; Hu et al. (2017) studied the impact of CQ on innovation performance [52]; and Golgeci et al. (2017) verified the mediating effect of CQ on the organization's ability to absorb innovation ability [53]. Studies on the factors that influence CQ show that CQ formation requires the learning of multicultural knowledge and cross-cultural experience and training [23]. This is a long-term and

continuous process. Therefore, we hypothesize that the impact of CQ on innovation behavior is sustainable. This leads to the following hypothesis.

**Hypothesis 1 (H1):** Employees' CQ have a positive impact on their sustainable innovation behavior.

## Cultural intelligence and knowledge sharing

Knowledge sharing refers to behavior where individuals help others or cooperate with others to solve problems and conceive new ideas, by transferring information [25]. The essence of knowledge sharing is the process of spreading and transforming it, enhancing the value of knowledge and generating innovation. Scholars have applied different perspectives to divide knowledge sharing into tacit/explicit knowledge sharing and general/key knowledge sharing [30]. The factors that influence knowledge sharing among employees in an organization are mainly personal characteristics, organizational culture, and organizational structure [26]. The subjective cognition, behavior, and motivation of individuals are important factors influencing knowledge sharing; these can determine the willingness of employees to share knowledge. Knowledge management researchers and enterprise practitioners have proposed that organization culture diversity has an important impact on knowledge sharing and the long-term success of organizational knowledge management activities [28]. CQ improves employees' cultural diversity adaptability and communication ability [22, 54], impacting internal knowledge transfer and cross-cultural learning [55, 56]. Therefore, we hypothsize that there is a close relationship between employee' cultural intelligence and knowledge sharing.

To clarify the concept of CQ, Ang (2007) divided CQ into four dimensions: metacognition, cognition, motivation, and behavior. Metacognition focuses on higher-order cognitive processes; Cognition reflects knowledge of the norms, practices and conventions in different cultures acquired from education and personal experiences; Motivation reflects the capability to direct attention and energy toward learning about and functioning in situations characterized by cultural differences; Behaviour reflects the capability to exhibit appropriate verbal and nonverbal actions when interacting with people from different cultures [33]. This dimensional division of CQ indicates that CQ represents the cognitive ability of employees to gain cultural knowledge, and their motivation and action to learn knowledge. This includes the cognition of knowledge sharing, knowledge sharing motivation, and behavior. Metacognition, cognition, and motivation can support and encourage knowledge learning and sharing among employees, whereas behavior reflects actions and the results of that knowledge sharing [56, 57]. Based on the analysis above, the following hypothesis is proposed.

**Hypothesis 2 (H2):** Employee' CQ has a positive influence on knowledge sharing.

## Knowledge sharing and sustainable innovative behavior

Knowledge is considered a core resource for an organization. One of the most important purposes of knowledge management is to create value for enterprise development through the exchange, application, and innovation of enterprise knowledge [58]. An increasing number of researchers have proposed that knowledge management is a key success factor for an organization in building competitive advantage, and knowledge sharing is an important way for members of an organization to acquire knowledge [59, 60]. Organizational innovation relies on enterprise knowledge, which is inseparable from active employee participation and knowledge sharing [61].

Social exchange theory (Homans,1958) proposes that social exchange is a process of interpersonal interaction, where individuals exchange important resources to maintain a good interpersonal relationship by considering the relationship between reward and contribution

[62]. Information transmission and learning among organization members is a kind of social exchange behavior, and is considered knowledge sharing [63]. Knowledge sharing facilitates mutual learning among members within the organization, facilitates the exchange of knowledge and experience among members of the organization, and creates conditions for the birth of new ideas or creativity. Empirical studies have also verified the impact of employee knowledge sharing on innovation behaviors [61, 64]. Based on the analysis above, the following hypothesis is proposed:

**Hypothesis 3 (H3):** Employees' knowledge sharing has a positive impact on their sustainable innovation behavior.

## Mediating role of knowledge sharing

CQ is a kind of cross-cultural interaction ability, with a direct effect on employees' innovation behavior; however, there may also be an intermediary mechanism between CQ and innovation behavior. The studies discussed above indicate that CQ positively affects knowledge sharing; knowledge sharing affects employees' sustainable innovation behaviors; and knowledge sharing is an intermediate variable in the process by which CQ influences sustainable innovation behaviors. When employees are in a multicultural environment, individual CQ can promote knowledge exchange and communication among employees, and supports cross-cultural interaction among members. This facilitates the ability of employees to learn from each other, discuss difficult problems, propose new ideas, or find new ways to solve problems [65]. These activities support innovation. Ratasuk et al. (2020) studied the impact of team CQ on team innovation performance, finding that knowledge sharing is a mediating variable between team CQ and team innovation performance [38]. Therefore, we hypothesize that CQ affects knowledge sharing, and knowledge sharing among individuals advances employees' sustainable innovation behavior. Based on the analysis above, the following hypothesis is proposed:

**Hypothesis 4 (H4):** Knowledge sharing plays a mediating role in the impact of employee cultural intelligence on sustainable innovation behavior.

## Moderating effect of organizational culture differences

Organizational cultural differences are the manifestation of the cultural diversity in an organization, and have a lasting impact on employees' attitudes and behaviors [66]. Shein (1990) defined culture as the assumptions, values, and behaviors shared by members. Culture distinguishes one group from another and emphasizes the differences between different cultures [67]. Hofstede (1984) proposes that culture is the "common psychological procedure" that distinguishes one kind of people from others [68]. Most comparative studies focusing on culture and cross-cultural management have focused on national, religious, and ethnic cultural differences, or individual demographic differences. They have not, however, fully considered group differences in organizational culture. In fact, organizational culture reflects the cultural characteristics of the country, region, and industry where the organization is located, and reflects many different sub-cultures (subcultures) within the organization, which may be fragmented, complementary, and even opposed [69].

Organizational culture refers to the common values, beliefs, or expectations within an enterprise. Organizational culture partly assimilates the value orientation, attitude, and code of conduct of employees from different cultural backgrounds, forming a converged value system [70]. However, due to the diversity of organizational culture, there can be a simultaneous convergence and alienation of organizational culture [71]. Organizational cultural differences represent the characteristics of a diverse organizational culture, which are manifested as the

differences in the attitudes and behaviors of the organization's members. These have a sustainable impact on the communication methods and work behaviors of employees.

Diversity climate theory and related studies have proposed that employee attitudes towards others depends on how organizations treat them, and the diversity climate of organizations cause an infectious process [72]. Organizational culture difference is a significant manifestation of organizational diversity, which affects organizational working climate and employee behavior. Trait activation theory (Tett,2003) emphasizes the matching of individual traits and organizational climate, highlighting that individual traits are activated when the organizational climate accepts those traits [49]. Employee innovation behavior and knowledge sharing motivation are influenced by the interaction between individual characteristics (such as CQ) and the organizational cultural environment. Therefore, based on diversity climate theory and trait activation theory, and the above analysis of organizational culture differences, we hypothesize that organizational culture differences, as an important factor influencing organizational climate, play a moderating role in the relationship between CQ and employees' sustainable innovation behavior, CQ, and knowledge sharing.

**Hypothesis 5 (H5):** Organizational cultural differences have a moderating effect on the relationship between CQ and employee knowledge sharing.

**Hypothesis 6 (H6):** Organizational culture differences have a moderating effect on the relationship between CQ and employees' sustainable innovation behavior.

## Conceptual model

This paper studies the influence of an employee's cultural intelligence on sustainable innovation behaviors, and analyzes the mediating effect of knowledge sharing and the moderating effect of organizational culture differences. Based on the above theoretical analysis and hypotheses, the following conceptual model is proposed (Fig 1).

## Research design

### Samples and data

Questionnaires were collected from Chinese employees in 31 enterprises across 9 industries, including manufacturing, the financial industry, education and training, and others. Due to the limitation of Covid-19, our questionnaire survey is conducted in three ways: one is to make an appointment and mail the paper questionnaire; the other is to send and collect the electronic version of the questionnaire by E-mail; the third is to use the "Questionnaire Star" research tool. The whole questionnaire survey period was 2 months. A total of 450 questionnaires were issued and 395 questionnaires were received in return. After eliminating invalid

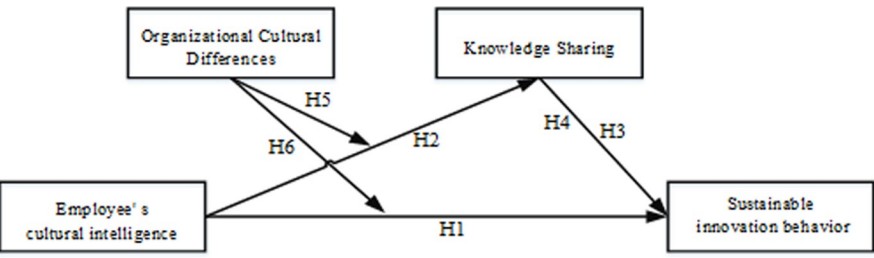

**Fig 1. Conceptual model.**

questionnaires with incomplete answers or identical answers across the survey, 336 valid questionnaires remained, making the effective rate of the questionnaire 74.66%.

Respondents were 50.89% male and 49.11% female; 64.29% of respondents reported holding a Bachelor degree; the others held a junior college degree or below, or graduate degree or above. The research object of this paper is the individual in the enterprise, and the subject are grass-roots employees, who are the front-line employees in the workplace, grass-roots staff accounted for 63.1% of respondents, grass-roots managers made up 19.64%, and the rest were middle and senior managers. In terms of experience, 19.35% of respondents had less than one year experience; 18.75% had 1–2 years; 22.32% had 3–5 years; 24.11% had 5–10 years; and 15.48% had more than 10 years experience. Respondents reported working in the areas of technical research, production and operation, marketing, finance, e-commerce, comprehensive management, and other areas.

## Measures

(1) **Cultural intelligence (CQ):** As noted above, Ang et al (2007). divided the CQ into metacognitiion, cognition, motivation, and behavior. This formed a classic four-dimensional Cultural Intelligence Theory [33], and resulted in a four-dimensional Cultural Intelligence Scale (CQS). So far, many studies have applied CQS to measure CQ. Bucker et al. analyzed the CQS proposed by Ang et al. They noted that the CQS did not assess the discriminant validity and correlation between the four dimensions, and the measurement results of the items caused multicollinearity among the four dimensions [73]. Bucker et al. applied the CQS using data for Chinese subjects with cross-cultural backgrounds, and deleted 8 items with poor factor loading in the CQS, further optimizing the scale. Bucker's scale was used in this study to measure the cultural intelligence of employees. The scale has 12 items and uses a Likert 7-point scale. Questions include: "I adjust my cultural knowledge as I interact with people from a culture that is unfamiliar to me" and "I vary the rate of my speaking when a cross-cultural situation requires it." The Cronbach's α of the original scale was 0.87.

(2) **Sustainable Innovation Behavior (SIB):** Individual innovation is the process of generating and implementing new ideas. Many scholars measure employees' innovation behaviors from the perspective of process [45, 74]. This study adopted the Innovation Behavior Scale developed by Scott and Bruce (1994), which is widely cited and shows high reliability. Based on the research background of sustainable innovation, this study modified the items, adding frequency adverbs such as "often," "constantly," and "usually" to reflect the sustainability of employees' innovation behavior. Sample items include: "I often come up with creative ideas" and "I usually find ways to get the resources I need to realize my creative ideas." The Cronbach's α of the original questionnaire was 0.916.

(3) **Knowledge Sharing (KS):** This study analyzed the willingness and behavior of organization members to share knowledge from the perspective of cross-cultural communication. As such, a scale developed by Faraj and Sproull (2000) was used for reference [75]. Srivastava et al. (2006) applied the scale in a study on employee relations, to measure the degree of knowledge sharing among organization members [76]. The questionnaire contains 4 items, such as: "The more knowledgeable members will provide the other members with knowledge or skills that are difficult to acquire for free" and "Members of the organization share their unique knowledge and expertise with each other." The Cronbach's α of the original questionnaire was 0.84.

(4) **Organizational Culture Differences (OCD):** Hofstede studied national cultural differences and proposed the classical cultural dimension theory. The theory addresses the dimensions and composition of corporate culture, by referring to national cultural differences. Organizational cultural differences are considered from three aspects: values, management behaviors, and systems [70]. Based on the research of Hofstede et al., Lavie et al. (2012) noted that differences in organizational culture mainly reflect differences in management style and organizational response, and developed a scale to capture these differences [77]. This study applied the scale developed by Lavie et al. [53]. Four questions measured differences in management styles, such as: "Organizations uses consensus seeking rather than authoritarian decision making." Five questions measured organizational responses, such as: "Organizations are open minded and creative in its approach to problem solving." The Cronbach's α of the original questionnaire was 0.78.

(5) **Control variables:** Based on previous studies, control variables included age, gender, education level, working years, and the positional level of employees [64]. Since there may be heterogeneity among employees in different industries and departments, we took the employee's industry as a control variable. These variables were set as dummy variables and were assigned values based on type and range. For example, "1" was set for male and "2" was set for female. The years of work experience were divided into "less than 1 year," "1–2 years," "3–5 years," and other levels. The education level variable was divided into junior college, undergraduate, and graduate degree and above.

## Reliability and validity analysis

**Confirmatory Factor Analysis (CFA).** In this study, the variables were measured using a maturity scale; an exploratory factor analysis are not required. Reliability and validity analyses were conducted using the measurement data. Variable verification factor analysis forms the basis for a latent variable analysis of a structural equation model [78]. The software package SPSS 23.0 was used for basic data processing. The Cronbach's α of the four variable items in the questionnaire were preliminarily calculated using a reliability analysis. The results showed that the Cronbach's α values of the items all exceeded 0.7. This indicated that the measurement items had a good explanatory ability. Cultural intelligence was assessed using 4 dimensions and 12 measurement items; as such, a second-order CFA was considered. In this study, Mplus7.4 is used for the confirmatory factor analysis.

The next step was a confirmatory factor analysis of the CQ. In the first-order CQ CFA model, the lowest correlation coefficient between the four dimensions was 0.47 (P<0.001). This indicated that the first-order factor analysis was suitable for an upgrade to the second-order CFA model. In the second-order model, the minimum factor loading of four dimensions was 0.755 (>0.7). This indicated that the second-order factor explained the variance with a higher first-order factor. Meanwhile, a comparison of the chi-square values ($\chi2$) of the first-order CFA and second-order CFA revealed that the target coefficient T value ($\chi2$ of the first-order model and $\chi2$ of the second-order model) was 94.35%. This indicated that the second-order model better explained the first-order model [79]. Based on the principle of simplification, the second-order CFA model was adopted to analyze the CQ. In the second-order CFA model, the items were modified to remove the two items affecting the residual independence and the measurement effect of the model due to the relatively low factor loading. Table 1 shows the specific analysis.

Table 1 shows that the factor loadings were all greater than 0.7, with a significant P-value. The combinatorial reliability (CR) of both the first-order model and the second-order model exceeded 0.7, showing good internal consistency and a good ability to interpret the

**Table 1. First order CFA/Second order CFA of CQ.**

| Model | Dime. | Item | factor loading | P-Value | CR | AVE | Fitting index |
|---|---|---|---|---|---|---|---|
| First order | MC | CQ1 | 0.871 | *** | 0.894 | 0.808 | $\chi^2$ = 71.792, RMSEA = 0.066 CFI = 0.981, TLI = 0.971 SRMR = 0.028 df = 29 |
| | | CQ2 | 0.926 | *** | | | |
| | COG | CQ4 | 0.787 | *** | 0.838 | 0.633 | |
| | | CQ5 | 0.850 | *** | | | |
| | | CQ7 | 0.746 | *** | | | |
| | MOT | CQ8 | 0.838 | *** | 0.861 | 0.756 | |
| | | CQ9 | 0.900 | *** | | | |
| | BEH | CQ10 | 0.849 | *** | 0.890 | 0.729 | |
| | | CQ11 | 0.874 | *** | | | |
| | | CQ12 | 0.838 | *** | | | |
| Second order | CQ | MC | 0.781 | *** | 0.900 | 0.693 | $\chi^2$ = 76.025, RMSEA = 0.066 CFI = 0.980, TLI = 0.971 SRMR = 0.030 df = 31 |
| | | COG | 0.755 | *** | | | |
| | | MOT | 0.924 | *** | | | |
| | | BEH | 0.859 | *** | | | |

Note: MC, COG, MOT, BEH represent the four dimensions of CQ respectively. MC = Metacognition, COG = Cognition, MOT = Motivation, BEH = Behaviour.
*** p<0.001

dimensions. The average variance extraction (AVE) represents the extraction amount of mean variance. The AVE exceeded 0.5; the dimension had a good average ability to explain the variables; and the fitting indexes of the model met the requirements.

To conduct a confirmatory factor analysis for other variables, there were 6 questions on employee sustainable innovation behavior (SIB), 4 questions on organizational knowledge sharing (KS), and 9 questions on organizational culture differences (OCD). The measurement model found that the standardized factor loadings of the 4 items related to knowledge sharing were high, and the model had a good degree of fit. Therefore, it was not necessary to modify the model. Factor loadings for 3 items related to employees' sustainable innovation behavior and organizational culture difference were lower than 0.6. The main fitting index of the model ($1<\chi^2/df<3$) was lower than 3. This indicated that the residual error values of some measurement items were not independent and were strongly correlated, and needed to be revised. Based on the modification index given by Mplus7.4 analysis, 2 items related to employees' sustainable innovation behavior and 4 items related to organizational culture differences were deleted. The modified factor loadings and reliability and validity analysis of the model are shown in Table 2 (including CQ).

The data analysis in Table 2 shows that the standardized estimate coefficients of the four variables' items and dimensions all exceeded 0.6; the standard error/estimate values were all greater than 1.96; and the P-value was less than 0.001. All the items and dimensions were significant. The item reliability is the square of the standardized estimate: a value greater than 0.36 is good and a value greater than 0.5 is very good. The analysis resulted in an R-square value greater than 0.36. The combined reliability (CR) of the four latent variables exceeded 0.7, showing good internal consistency and indicating that the dimensions had a good interpretation ability. The convergence validity (AVE) was greater than 0.5, and the average explanatory ability of the dimension was good.

**Discriminant validity.** According to Fornellr and Larcker (1981), the triangle below the diagonal of the correlation coefficient matrix of variables represents the square root of AVE [80]. The values of the square root of AVE were compared with the coefficients of other related

**Table 2. CR & AVE.**

| Dim. | Item | Parameters of the significance test | | | | Item Reliability | Composite Reliability | Convergence Validity |
|---|---|---|---|---|---|---|---|---|
| | | Estimate | S.E. | Est./S.E. | P-Value | R-SQUARE | CR | AVE |
| SIB | SIB1 | 0.707 | 0.031 | 22.485 | *** | 0.500 | 0.867 | 0.621 |
| | SIB2 | 0.903 | 0.019 | 46.768 | *** | 0.815 | | |
| | SIB3 | 0.775 | 0.027 | 29.146 | *** | 0.601 | | |
| | SIB6 | 0.754 | 0.028 | 26.730 | *** | 0.569 | | |
| KS | KS1 | 0.794 | 0.024 | 32.646 | *** | 0.630 | 0.876 | 0.639 |
| | KS2 | 0.820 | 0.022 | 37.125 | *** | 0.672 | | |
| | KS3 | 0.880 | 0.018 | 49.409 | *** | 0.684 | | |
| | KS4 | 0.827 | 0.022 | 38.458 | *** | 0.569 | | |
| OCD | OCD2 | 0.677 | 0.034 | 20.187 | *** | 0.458 | | |
| | OCD4 | 0.707 | 0.031 | 22.541 | *** | 0.500 | | |
| | OCD6 | 0.864 | 0.020 | 42.890 | *** | 0.746 | 0.876 | 0.587 |
| | OCD7 | 0.802 | 0.025 | 32.647 | *** | 0.643 | | |
| | OCD9 | 0.765 | 0.027 | 28.267 | *** | 0.585 | | |
| CQ | MC | 0.781 | 0.031 | 25.252 | *** | 0.781 | 0.900 | 0.693 |
| | COG | 0.755 | 0.035 | 21.561 | *** | 0.755 | | |
| | MOT | 0.924 | 0.024 | 38.959 | *** | 0.924 | | |
| | BEH | 0.859 | 0.026 | 33.259 | *** | 0.859 | | |

***p<0.001

dimensions. Table 3 shows that the square root of AVE was greater than the correlation coefficient of other dimensions. This indicated there was significant discriminant validity between the model dimensions.

## Model analysis

### Basic structure equation model analysis

Mplus7.4 was used to conduct a regression analysis on the basic structural equation model of the conceptual model (without considering the moderator variable). Fig 2 shows the results and indicate that, without considering the moderating effect of organizational cultural differences (OCD), employees' CQ significantly influenced sustainable innovation behavior (SIB)

**Table 3. Differential validity.**

| Latent variable | AVE | SIB | KS | OCD | MC | COG | MOT | BEH | CQ |
|---|---|---|---|---|---|---|---|---|---|
| SIB | 0.621 | **0.788** | | | | | | | |
| KS | 0.639 | 0.546 | **0.799** | | | | | | |
| OCD | 0.587 | 0.519 | 0.711 | **0.766** | | | | | |
| MC | 0.808 | 0.607 | 0.578 | 0.476 | **0.899** | | | | |
| COG | 0.633 | 0.565 | 0.539 | 0.443 | 0.594 | **0.796** | | | |
| MOT | 0.756 | 0.674 | 0.643 | 0.529 | 0.709 | 0.660 | **0.869** | | |
| BEH | 0.729 | 0.669 | 0.638 | 0.525 | 0.704 | 0.655 | 0.782 | **0.854** | |
| CQ | 0.693 | 0.759 | 0.724 | 0.596 | 0.799 | 0.744 | 0.888 | 0.881 | **0.832** |

Note: The diagonal boldface is the square root of AVE, and the lower triangle is the Pearson correlation of the dimensions.

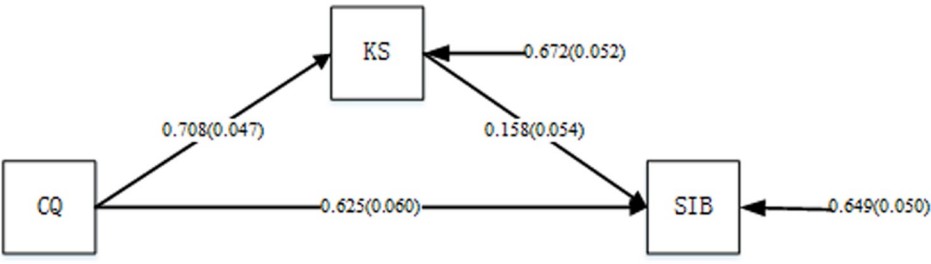

**Fig 2. The basic structure equation model analysis.**

(r = 0.625, P < 0.001). Individual cultural intelligence also affected knowledge sharing (r = 0.708, P < 0.001). At the same time, knowledge sharing (KS) also affected sustainable innovation behavior (r = 0.158, P < 0.05). These results verified Hypothesis 1, Hypothesis 2, and Hypothesis 3. Meanwhile, according Baron to and Kenny's (1986) analysis of mediating effect [81], knowledge sharing is initially considered to be an intermediary factor between individual cultural intelligence and sustainable innovation behavior.

## Path analysis

Driven by the basic model analysis, the overall structural equation model (SEM) was analyzed to evaluate the moderating effect of organizational culture differences. The SEM analysis found that the fitting degree was good ($\chi 2$/df = 1.954, CFI = 0.955, TLI = 0.948, RMSEA = 0.054, SRMR = 0.043). Table 4 shows the specific model fitting results. The results indicated that the fitting index of the SEM conformed to the recommended value, and there was only a small difference between the sample and the model. As such, the model has a high suitability.

Table 5 shows the path analysis of the SEM and the correlation of all variables after adding the moderator variable. The impact of employee cultural intelligence on sustainable innovation behavior was significant (r = 0.593, P<0.001); the impact of cultural intelligence on knowledge sharing was significant (r = 0.462, P<0.001); and the impact of knowledge sharing on employee sustainable innovation behavior was significant (r = 0.103, P<0.05). These results again verified Hypothesis 1, Hypothesis 2, and Hypothesis 3. The interaction between cultural intelligence and organizational culture difference (CQ×OCD) significantly influenced knowledge sharing (r = -0.119, P<0.05), indicating that organizational culture difference had a moderating effect on the relationship between CQ and employees' knowledge sharing behavior. This verified Hypothesis 5. The interaction between cultural intelligence and organizational culture difference (CQ×OCD) had no significant influence on employees' sustainable

**Table 4. Fitting index of overall SEM.**

| Fitting index | Key value | Model indexes | Fit |
|---|---|---|---|
| ML $\chi^2$ | The smaller, the better | 429.946 | |
| Df | The bigger, the better | 220 | |
| $\chi^2$/Df | 1<$\chi^2$/Df<3 | 1.954 | Fit |
| CFI | >0.9 | 0.955 | Fit |
| TLI | >0.9 | 0.948 | Fit |
| RMSEA | <0.08 | 0.054 | Fit |
| SRMR | <0.08 | 0.043 | Fit |

**Table 5. SEM path analysis.**

| Path | Parameters of the significance test | | | | Hypothesis |
|---|---|---|---|---|---|
| | Estimate | S.E. | Est./S.E. | P-Value | |
| CQ→KS | 0.462 | 0.051 | 9.044 | *** | Support |
| OCD→KS | 0.386 | 0.042 | 9.191 | *** | Support |
| CQ→SIB | 0.593 | 0.062 | 9.564 | *** | Support |
| KS→SIB | 0.103 | 0.050 | 2.060 | ** | Support |
| OCD→SIB | 0.120 | 0.052 | 2.307 | ** | Support |
| CQ × OCD→KS | -0.119 | 0.066 | -1.803 | ** | Support |
| CQ × OCD→SIB | -0.004 | 0.076 | -0.052 | 0.959 | Not support |

**p<0.05,

***p<0.001

innovation behavior (R = -0.004, P = 0.959). This indicated that organizational culture difference had no moderating effect on the relationship between CQ and employees' sustainable innovation behavior. As a result, Hypothesis 6 was not supported.

## Mediating effect test

The classic mediating effect test method is the stepwise method proposed by Baron and Kenny (1986) [81]; however, this method does not allow the observation of the neutralization and masking effects of mediating variables. The Sobel test assumes the normal distribution of interaction terms; however, the product (interaction) of two coefficients is usually not normal, causing that test to be questioned. Currently, the Bootstrap method with bias correction is more popular, with an increasing number of scholars accepting that it has higher testing power and more accurate confidence interval [82]. The bootstrap method in Mplus7.4 was used to further test the mediating effect of knowledge sharing. Table 6 shows the test results and indicates that the path coefficient of the mediating effect of knowledge sharing between employee CQ and sustainable innovation behavior is 0.100; the 95% CI (confidence interval) of the deviation correction is = [0.012,0.188]. The upper and lower limits of the confidence intervals don't include 0, demonstrating the presence of an indirect mediating effect. This supports Hypothesis 4. Meanwhile, Table 6 compares the mediating effect, total effect, and the influencing effect of CQ on sustainable innovation behavior.

**Table 6. The mediating test of knowledge sharing.**

| Effect | Point Estimate | Product of Coefficients | | | Bootstrap 1000 Times 95% CI | | | |
|---|---|---|---|---|---|---|---|---|
| | | | | | Percentile | | Bias corrected percentile | |
| | | S.E. | Est./S. | P-Value | Lower | Upper | Lower | Upper |
| CQ→SIB | 0.616 | 0.067 | 9.194 | *** | 0.487 | 0.729 | 0.485 | 0.745 |
| CQ→KS→SIB | 0.100 | 0.045 | 2.227 | ** | 0.008 | 0.185 | 0.012 | 0.18 |
| TOTAL | 0.716 | 0.058 | 12.342 | *** | 0.603 | 0.823 | 0.602 | 0.822 |

**p<0.05,

***p<0.001

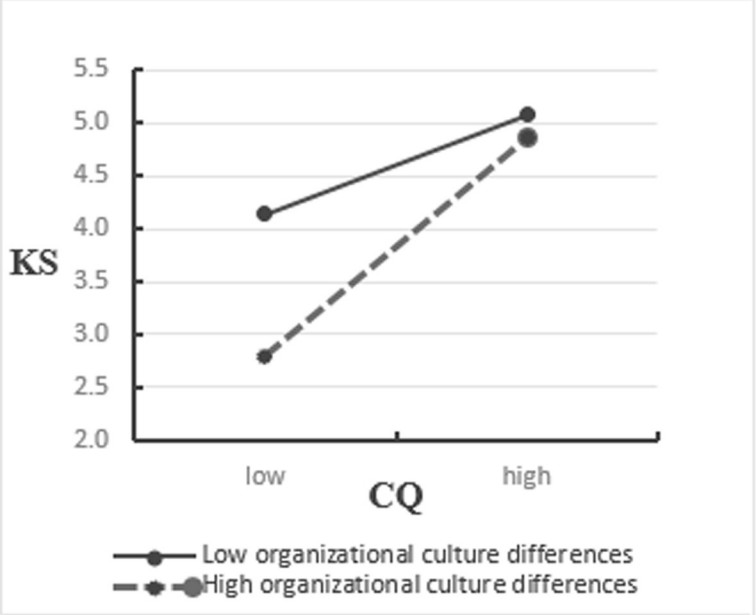

**Fig 3. Test of the moderating effect of organizational cultural differences on CQ and knowledge sharing.**

## Moderating effect test

The path analysis of the SEM verified Hypothesis 5. Organizational culture differences have a moderating effect on the relationship between CQ and employees' knowledge sharing behavior. Hypothesis 6 was not supported. Organizational cultural differences had no moderating effect on the relationship between CQ and employees' innovation behaviors (Table 5). To further analyze Hypothesis 5, the interaction between CQ and organizational culture difference was decomposed based on the study of Aiken and West (1991) [83], and a slope analysis was conducted using the limits of one standard deviation above and one standard deviation below the variable mean. These analyses illustrated the impact of CQ on knowledge sharing when there are organizational culture differences (**Fig 3**). The figure shows that the organizational culture difference had a clear moderating effect.

## Conclusion and discussion

### Conclusion

This study examined the impact of employee cultural intelligence on sustainable innovation behavior in the context of organizational cultural diversity. Knowledge sharing was established as the intermediary variable and organizational cultural difference was established as the moderator variable to explain the mechanism of action between them. The results showed six key findings. (1) Employees' cultural intelligence had a significant positive impact on sustainable innovation behavior. (2) Employees' cultural intelligence significantly positively impacted sustainable innovation behavior. (3) Employees' cultural intelligence positively affected employees' knowledge sharing behavior. (4) Knowledge sharing played a partial mediating role in the positive relationship between employees' cultural intelligence and sustainable innovation behavior. (5) Organizational culture differences negatively regulated the relationship between CQ and knowledge sharing. When organizational culture differences were small, CQ had a

stronger influence on knowledge sharing. (6) The moderating effect of organizational culture difference on CQ and employees' sustainable innovation behavior was not significant.

In the context of organizational cultural diversity, employees with high CQ, such as those with strong cross-cultural management and communication skills, are more likely to generate innovative ideas and put them into action. Employees with high CQ are more likely to share knowledge with others, and those knowledge sharing behaviors further promote the generation of innovative ideas and employee behavior. In the context of cultural diversity, cultural consistency and cultural difference coexist within an organization. When the consistency of organizational culture is strong, the cultural difference is small and employees with a high CQ are more likely to share knowledge with others. This facilitates innovation. In contrast, when cultural differences become larger, knowledge sharing may be weakened. Therefore, the interaction between organizational cultural differences and knowledge sharing may negatively affect sustainable innovation behaviors. These theoretical results are consistent with the management practice of strengthening corporate culture construction and unifying values to promote corporate development.

It is important to examine Hypothesis 6, which was not supported. Organizational culture difference was found to have no significant moderating effect on the relationship between employees' CQ and sustainable innovation behavior. Employee CQ was found to promote sustainable innovation behavior, however, the interaction between organizational culture differences and CQ had no significant relationship with sustainable innovation behavior. These results showed that the relationship between CQ and sustainable innovation was not sensitive to the change of organizational culture differences. This may be because the sustainable innovation behaviors of employees reporting a high CQ were not directly related to the differences of organizational culture. Employees with high CQ reported having strong cultural metacognition, understanding more diverse cultural knowledge, having strong cross-cultural motivation and, take the initiative to adjust cross-cultural behaviors. Employees with high CQ reported being more likely to adapt to cultural diversity and were less sensitive to cultural differences. This may also be because of the complementary effect between CQ and organizational culture differences, resulting in no significant relationship with innovation behavior.

## Theoretical significance

(1) This study expanded our understanding of antecedent variables and the production mechanism of employees' sustainable innovation behavior. Previous studies have paid less attention to the individual's cultural diversity management ability. Therefore, CQ was used as an antecedent variable to measure the communication and management performance of employees in different organizational cultural environments. The study also incorporated the hard-to-measure influencing factor of organizational cultural diversity into the analytical framework of individual sustainable innovation behaviors. The study of individual innovation, from the perspective of organizational culture management, provides a new way to study employees' sustainable innovation behavior.

(2) This study enriches the existing research on CQ theory. Most studies on CQ theory have focused on its influencing factors and have discussed its antecedent variables, such as the Big Five personality, overseas experience, and employee training [21, 23, 84]. Most studies examining the effect of CQ have focused on job performance, job adjustment, and cross-text adaptability [23, 34]. Few have evaluated sustainable innovation. Therefore, this research enriches the research field of CQ theory.

(3) This paper combined cultural intelligence theory, self-determination theory, social exchange theory, and organizational diversity climate theory to study employees' sustainable

innovation behavior. Adding knowledge-sharing factors captured the interaction between individuals and organizations as mediating variables, and considered the moderating effect of organizational cultural differences. We integrated several theories related to innovation behavior into a unified analytical framework and realized the integration of theories.

## Administration recommendations

The results above highlight several recommendations for consideration. First, it is important to strengthen the management of cultural diversity in organizations. The diversity of organizational culture is an important intangible resource of an enterprise, and has a profound impact on knowledge management and sustainable innovation. However, organizational culture diversity is a double-edged sword. Some managers hold that diversity brings innovation and high performance [2, 85], and maximize cultural diversity in management practices (such as human resource recruitment). Some studies have shown that cultural diversity can lead to discrimination, conflicts, and differences, and eventually may become a cultural barrier to sustainable development of organizations, therefore, consistency of organizational culture should be advocated in management practice.

This study argues that it is more meaningful to discuss how to do a good job in managing cultural diversity, rather than arguing about the pros and cons of cultural diversity. Therefore, this paper discusses the influence of individual cross-cultural management ability (CQ) on sustainable innovation behaviors, and calls on managers to strengthen cross-cultural management. Through the research, we found that the organizational culture of some small and medium-sized enterprises is somewhat spontaneous and fuzzy, and some of the newest generation of employees are personalized [3, 86]. This can lead to more cultural diversity within the organization. However, the lack of management of cultural diversity in the enterprise has formed cultural barriers, affecting employee innovation and development. Some other surveyed companies, such as Huawei and Haier, have attached importance to cultural diversity management and cultural intelligence training, and their employees have been reported to perform better in sustainable innovation [87, 88]. Cultural diversity management supports sustainable innovation. Managers should improve employees' CQ through cross-cultural experience and training, and control the level of organizational cultural differences.

The next recommendation is to improve employee cultural intelligence to achieve higher performance. Cultural intelligence promotes innovative behavior and positively impacts employee performance and job adaptation. Employees with high cultural intelligence may have better communication skills with colleagues from different cultural backgrounds, may be more adaptable, and may be able to better manage the organization's multicultural environment. These are important employee characteristics and qualities. Therefore, in the process of human resource recruitment, organizations should pay attention to cultural intelligence capabilities and select staff with high cultural intelligence who can effectively integrate into the corporate culture. In the field of human resources training, we should pay attention to cultivating employees' cognitive abilities with respect to multi-culturalism, increase education about cross-cultural knowledge, and increase the use of cross-cultural experience projects to enhance cultural intelligence.

The final recommendation is to improve the level of employee knowledge sharing. The nature of knowledge sharing is interaction and learning, and while knowledge sharing reflects an individual's behavior, it also forms an organizational climate. If the staff in the organization are willing to share knowledge with others and learn from each other, it forms a transmission mechanism that promotes innovative behavior. Therefore, managers should promote

knowledge-sharing behavior within the organization. This includes strengthening support for knowledge-sharing behavior in performance appraisals, organizational culture, and system construction, including through targeted performance awards, organized results sharing and seminars, and the award of honorary titles.

## Research limitations and prospects

There were three main limitations in this study, which highlight opportunities for future research. First, there were limitations to the research sample. Study subjects represented different industries, including education and training, manufacturing, finance, and e-commerce, and reported engaging in different types of work. This leads to the advantage of having results that are more universal; however, the study was not representative of a specific industry and occupation. Future research could discuss specific industries or professions, such as finance or technology, and discuss the impact of CQ on more specific sustainable innovation behaviors.

Second, there were limitations in the study's measurement scale. This study did not develop a new scale; rather, it used mature scales. The advantage of this approach was that the scale had a high level of reliability; however, this study involved organizational culture, and different countries and regions have different organizational culture backgrounds. As such, the scale may be biased. Future studies could avoid this problem by combining multiple research methods and collecting data across multiple time stages.

Third, there were some limitations in research content. CQ is related to emotional intelligence (EQ), general intelligence (IQ), self-efficacy, and other factors. However, these were not studied in detail due to space and research model limitations. This may lead to some confusion, and deserves future study. In addition, the relationship between organizational cultural intelligence, organizational knowledge sharing climate, and organizational sustainable innovation remains to be explored. These are areas for future research.

Despite these limitations, this study provided valuable insights about theoretical guidance for managing organizational cultural diversity and advancing cultural intelligence and sustainable innovation behaviors among employees. At the same time, we emphasized that more attention should be paid to the management of cultural diversity and the sustainable behavior of employees in the management practice.

## Supporting information

**S1 Appendix. Questionnaire.**
(DOCX)

**S1 File. Relevant data underlying the findings described in manuscript.**
(XLS)

**S2 File. Means values and std.**
(DOCX)

## Acknowledgments

LJL would like to thank Prof. Xiong Shengxu for discussing the issue of individual CQ, which inspired us a lot. Thanks to several colleagues for their help in data collection and data analysis. We are grateful to three anonymous reviewers and our academic editor, for their thoughtful critiques and suggestions.

## Author Contributions

**Conceptualization:** Jinlong Li.

**Formal analysis:** Jinlong Li, Shengxu Xiong.

**Funding acquisition:** Shengxu Xiong.

**Investigation:** Na Wu, Shengxu Xiong.

**Methodology:** Jinlong Li.

**Resources:** Na Wu, Shengxu Xiong.

**Software:** Jinlong Li, Na Wu.

**Writing – original draft:** Jinlong Li.

**Writing – review & editing:** Jinlong Li.

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
