## [Decision Letter · Decision Letter 0]

25 Jan 2021

PONE-D-20-36985

Sustainable Innovation in the context of organizational Cultural Diversity: The role of Cultural Intelligence and knowledge sharing

PLOS ONE

Dear Dr. Jinlong,

Thank you for submitting your manuscript to PLOS ONE. After careful consideration, we feel that it has merit but does not fully meet PLOS ONE’s publication criteria as it currently stands. Therefore, we invite you to submit a revised version of the manuscript that addresses the points raised during the review process.

We look forward to receiving your revised manuscript.

Kind regards,

Nadja Damij, Ph.D.

Academic Editor

PLOS ONE

Additional Editor Comments:

Dear Authors,

Kindly review your manuscript based on the reviews below.

Best regards,

Dr. Nadja Damij

Journal Requirements:

2.) Thank you for stating the following in the Funding Section of your manuscript:

'This study was supported by grants from the National Social Science Foundation of China (Project No. 15BGL036), the Innovation Program of Zhongnan University of Economics and Law(Project No. 201911049).'

'This paper was written by Li Jinlong . in collaboration with all co-authors. The formal were analyzed by Li Jinlong and Wu Na.and Xiong Shengxu.The funding was acquisited by Xiong Shengxu. Data was calculated by Li Jinlong and Wu Na.'

3.) In your Data Availability statement, you have not specified where the minimal data set underlying the results described in your manuscript can be found. PLOS defines a study's minimal data set as the underlying data used to reach the conclusions drawn in the manuscript and any additional data required to replicate the reported study findings in their entirety. All PLOS journals require that the minimal data set be made fully available. For more information about our data policy, please see http://journals.plos.org/plosone/s/data-availability.

Reviewers' comments:

Reviewer's Responses to Questions

**Comments to the Author**

1. Is the manuscript technically sound, and do the data support the conclusions?

Reviewer #1: Yes

Reviewer #2: Yes

Reviewer #3: Yes

2. Has the statistical analysis been performed appropriately and rigorously? 

Reviewer #1: Yes

Reviewer #2: Yes

Reviewer #3: Yes

3. Have the authors made all data underlying the findings in their manuscript fully available?

Reviewer #1: Yes

Reviewer #2: Yes

Reviewer #3: Yes

4. Is the manuscript presented in an intelligible fashion and written in standard English?

Reviewer #1: Yes

Reviewer #2: Yes

Reviewer #3: Yes

5. Review Comments to the Author

Reviewer #1: Very well written, sound theoretical background and appropriate methodological approach. You can expand on sampling strategy used, period of data collection, but also how exactly were the respondents approached due to COVID-19 restrictions and working from home rules. Please also add a definition of the term grass-root employee. I believe there is high heterogeneity between different sectors which can be added as a limitation.

Reviewer #2: 1. Overall, the paper is well-written with good readability and organisation. A few minor changes are suggested below, which could improve the work further.

2. The paper has frequently used the term ‘sustainable innovation behavior’. Sustainable innovation introduces the concept of sustainability in the innovation process in which typically the environment, social and financial considerations are paid due attention. Presumably, in this paper, the authors have used the term to indicate the ‘continuous (or sustained) innovation behavior’ (line 113) of the employees. If that’s the case, to avoid confusion, it is suggested that the term ‘sustained’ be either omitted or be replaced by something more specific (for example: continuous or sustained).

3. It will be useful to include the survey questions in the paper as supporting information.

4. The study involves survey participants from 31 organisations across several industry sectors. Future research to investigate cultural diversity/intelligence variations among the industries and/or geographies would be exciting.

Reviewer #3: Jinlong Li et al. manuscript “Sustainable innovation in the context of organizational cultural diversity: the role of cultural intelligence and knowledge sharing” focuses on issues that are current and significant from a theoretical and practical point of view, namely the impact of employees’ cultural intelligence on sustainable innovative behavior in the context of organizational cultural diversity as a dominant phenomenon in the modern global economy.

The article is written in accordance with established academic standards for this type of study. According to the authors, with whom I agree, the following key results have been obtained on the basis of theoretical assumptions and their own empirical study, of which I will mention the most important ones: employee cultural intelligence has a significant positive impact on their sustainable innovative behavior and has a positive impact on employee knowledge sharing. Knowledge sharing, in turn, plays a partial mediating role in establishing a positive relationship between employees' cultural intelligence and sustainable innovative behavior. At the same time, organizational cultural differences have been shown to negatively regulate the relationship between employee cultural intelligence and knowledge sharing. It was also shown, contrary to the authors' expectation, that organizational cultural differences do not have a significant moderating effect on the relationship between employees' cultural intelligence and their sustainable innovative behavior.

Based on the aforementioned results, the manuscript has both theoretical and practical significance. Namely, the study expands our understanding of antecedent variables and mechanisms of sustainably innovative employee behavior, which in the modern economy brings an important comparative advantage to companies competing in the global markets. The study also enriches the existing research field of cultural intelligence theory by integrating it with self-determination theory, social exchange theory and organizational climate diversity theory in order to capture innovative behavior in a common analytical and theoretical framework.

The achievements of this study may also have practical relevance, especially in raising the level of knowledge sharing among employees, by improving the cultural intelligence of employees in order to achieve higher performance and in raising awareness of the importance of managing intercultural differences in the organization. Uncontrolled cultural differences can lead to disputes, conflicts, discrimination and thus become a cultural barrier to sustainable development of the company. Finally, the authors argue one important point of view in discussions on this topic: it makes more sense to discuss how to do a good job in managing cultural differences than to engage in empty debates pro et contra cultural differences.

The study has some limitations. The authors are aware of at least three of them: sample limitations, measurement scales, and content. I would add another to this, namely the problem of overgeneralization in the formulation of hypotheses. Let us examine one example. "H1: Employee' CQ has a positive influence on knowledge sharing." What, how much, in what way, etc.? A hypothesis that is adequately logically formulated should be more specific. A commonplace in scientific methodology is that a hypothesis is verified based on successful predictions. Therefore, its specific/concrete formulation is important, which is not the case with all 6 hypotheses stated by the authors. If a hypothesis does not explicitly or implicitly demarcate what it is specifying, then it cannot be taken as adequate. Most importantly, there must be a way to reject the hypothesis. For this reason, I would advise the authors to make an additional effort before the manuscript goes to publication to reframe their basically good hypotheses to be more precisely worded.

Finally, despite these limitations, I would argue that the manuscript offers valuable insights and that due to its intellectual interest will influence future research on topics such as management of intercultural differences, organizational climate of knowledge sharing, cultural intelligence and sustainable innovative behavior of employees.

6. PLOS authors have the option to publish the peer review history of their article (what does this mean?). If published, this will include your full peer review and any attached files.

Reviewer #1: No

Reviewer #2: No

Reviewer #3: No

---

## [Author Response · Author response to Decision Letter 0]

25 Feb 2021

Reply to the first reviewer

Dear reviewer:

Thank you very much for your affirmation, which is the greatest encouragement to us!

1.Due to the limitation of Covid-19, our questionnaire survey is conducted in three ways: one is to make an appointment and mail the paper questionnaire; the other is to send and collect the electronic version of the questionnaire by E-mail; the third is to use the "Questionnaire Star" research tool. The whole questionnaire survey period was 2 months. 

2.As for grass-roots employees, the research object of this paper is the individual in the enterprise, and the subject is grass-roots employees, who are the front-line employees in the workplace, accounting for 63.1%. I strongly agree with you that there is a high degree of heterogeneity between different departments. For example, the CQ and innovation of employees in R&D and sales departments should be different. However, the survey subjects in this paper do not distinguish departments, which is the research limitation of this paper. We will be paid attention to in the later research. I agree to use the industry (department) of the employee as the control variable.

Thank you very much.

With kind regards

Li jinlong Wu na Xiong shengxu

Reply to the second reviewer:

Dear reviewer:

Thank you very much for the affirmation of the article. We accept your opinions and will continue to improve.

1.We agree with your understanding of the term sustainable innovation behavior.To avoid confusion, we accept your suggestion and using “sustained” instead. Avoid confusion between the section you pointed out (line 113) and the explanation that follows.

2. Based on your comments, we will provide a questionnaire as supplementary information.

3. Thank you for your guidance for future research. Based on your comments, the authors of this paper plan to do further in-depth research into the differences in CQ in specific industries or in specific regions.

Thank you very much.

With kind regards

Li jinlong Wu na Xiong shengxu

Reply to the third reviewer:

Dear reviewer:

Thank you for your careful reading and objective evaluation,we are encouraged!

We humbly accept the hypothesis analysis problem you raised and take this problem as the research limitation of this paper. However, due to the limitation of theoretical level and language, it is difficult to revise all the assumptions. We will try our best to revise them before the publication of the manuscript. We also look forward to your further help. We will avoid this question in later writing. In addition, before the manuscript is published, we will make more efforts to make the statement more clear and try our best to do better.

Thank you very much.

With kind regards

Li jinlong Wu na Xiong shengxu

---

## [Decision Letter · Decision Letter 1]

16 Apr 2021

Sustainable Innovation in the context of organizational Cultural Diversity: The role of Cultural Intelligence and knowledge sharing

PONE-D-20-36985R1

Dear Dr. Jinlong,

We’re pleased to inform you that your manuscript has been judged scientifically suitable for publication and will be formally accepted for publication once it meets all outstanding technical requirements.

Kind regards,

Nadja Damij, Ph.D.

Academic Editor

PLOS ONE

Additional Editor Comments (optional):

Reviewers' comments:

Reviewer's Responses to Questions

**Comments to the Author**

1. If the authors have adequately addressed your comments raised in a previous round of review and you feel that this manuscript is now acceptable for publication, you may indicate that here to bypass the “Comments to the Author” section, enter your conflict of interest statement in the “Confidential to Editor” section, and submit your "Accept" recommendation.

Reviewer #1: All comments have been addressed

Reviewer #2: (No Response)

Reviewer #3: All comments have been addressed

2. Is the manuscript technically sound, and do the data support the conclusions?

Reviewer #1: Yes

Reviewer #2: Partly

Reviewer #3: Yes

3. Has the statistical analysis been performed appropriately and rigorously? 

Reviewer #1: Yes

Reviewer #2: I Don't Know

Reviewer #3: Yes

4. Have the authors made all data underlying the findings in their manuscript fully available?

Reviewer #1: Yes

Reviewer #2: Yes

Reviewer #3: Yes

5. Is the manuscript presented in an intelligible fashion and written in standard English?

Reviewer #1: Yes

Reviewer #2: Yes

Reviewer #3: Yes

6. Review Comments to the Author

Reviewer #1: Thank you for providing further clarifications regarding the comments I have previously raised. This meets my expectations and I my opinion adds well to the overall structure of the paper. Wish you good luck with the publication and hope the readers will enjoy the paper.

Reviewer #2: Thanks for addressing the comments raised by the reviewer. The survey questionnaire has been provided as supporting information with the revised manuscript. Given that the entire survey was conducted online (via direct email or using a survey tool), the questionnaire content, in which survey participants were asked to respond to the questions with numeric choices (1 through 7), looks a bit ambiguous. There is no instruction in the questionnaire to suggest what a numeric value (across the range of 1 to 7), actually means to the survey respondent and, accordingly, is subject to an individual's interpretation.

Reviewer #3: (No Response)

7. PLOS authors have the option to publish the peer review history of their article (what does this mean?). If published, this will include your full peer review and any attached files.

Reviewer #1: No

Reviewer #2: No

Reviewer #3: No

---

## [Editor Report · Acceptance letter]

22 Apr 2021

PONE-D-20-36985R1 

Sustainable Innovation in the context of organizational Cultural Diversity: The role of  Cultural Intelligence and knowledge sharing 

Dear Dr. Li:

I'm pleased to inform you that your manuscript has been deemed suitable for publication in PLOS ONE. Congratulations! Your manuscript is now with our production department. 

Kind regards, 

on behalf of

Dr. Nadja Damij 

Academic Editor

PLOS ONE